# Studies on Chemical IR Images of Poly(hydroxybutyrate–*co*–hydroxyhexanoate)/Poly(ethylene glycol) Blends and Two-Dimensional Correlation Spectroscopy

**DOI:** 10.3390/polym11030507

**Published:** 2019-03-17

**Authors:** Yeonju Park, Sila Jin, Yujeong Park, Soo Min Kim, Isao Noda, Boknam Chae, Young Mee Jung

**Affiliations:** 1Department of Chemistry, Institute for Molecular Science and Fusion Technology, Kangwon National University, Chuncheon 24341, Korea; yeonju4453@kangwon.ac.kr (Y.P.); jsira@kangwon.ac.kr (S.J.); po01266@naver.com (Y.P.); italy3255@naver.com (S.M.K.); 2Department of Materials Science and Engineering, University of Delaware, Newark, DE 19716, USA; 3Danimer Scientific, 140 Industrial Blvd., Bainbridge, GA 39817, USA; 4Pohang Accelerator Laboratory, POSTECH, Pohang 37673, Korea; cbn@postech.ac.kr

**Keywords:** biodegradable polymer, IR imaging, 2D-COS, blend

## Abstract

Biodegradable poly-[(R)-3-hydroxybutyrate-*co*-(R)-3-hydroxyhexanoates] (PHBHx) have been widely studied for their applications in potentially replacing petroleum-based thermoplastics. In this study, the effect of the high molecular weight (*M*n = 3400) poly(ethylene glycol) (PEG) blended in the films of PHBHx with different ratios of PEG was investigated using chemical FTIR imaging. Chemical IR images and FTIR spectra measured with increasing temperature revealed that PEG plays an important role in changing the kinetics of PHBHx crystallization. In addition, two-dimensional correlation spectra clearly showed that thermal properties of PHBHx/PEG blend film changed when the blending ratio of PHBHx/PEG were 60/40 and 50/50. Consequently, PEG leads to changes in the thermal behavior of PHBHx copolymers.

## 1. Introduction

Environmental pollution has been a serious issue over the last few decades. Plastic pollution especially is a huge problem. Because the degradation rate of plastic is much slower than that of production, and its durability rate and recycled portion is as low as 9% [1,2], it is constantly accumulating on the earth. There have been many efforts to reduce plastic consumption or promote plastic recycling, yet only some have had limited success so far. Therefore, the development of easily degradable polymers has received significant interest [3,4] as an alternative approach.

Poly(hydroxyalkanoates) (PHAs) produced by a variety of bacteria are well-known biodegradable polymers [5,6,7]. Among them, poly-[(R)-3-hydroxybutyrate) (PHB) is one of the well-studied biodegradable polymers due to its potential to replace petroleum-based thermoplastics [5]. However, thermal, mechanical and chemical properties of PHB homopolymer by itself are somewhat limited for wider applications. To improve the properties of PHB, many attempts have been studied by copolymerization [2,8,9,10,11,12] and blending [13,14,15] with the other polymers. The PHB-based copolymers have been extensively studied using infrared (IR) [16,17,18,19] and Raman [8,10,20] spectroscopies, wide-angle X-ray scattering (WAXS) [11,20,21], X-ray diffraction (XRD) [22,23], differential scanning calorimetry (DSC) [24,25], and so on.

As polymer systems are very complicated to investigate using only conventional spectroscopic analysis, many kinds of advanced analytical methods, such as the secondary derivative, chemometrics, and two-dimensional correlation spectroscopy (2D-COS) analysis have been used. Among them, 2D-COS is a very powerful and reliable analytical method for polymer study [26,27,28,29,30,31,32,33,34]. 2D-COS can sort out very important and interesting information in the system, which is sometimes scarcely detected in conventional 1D spectral analysis. It has many powerful advantages to greatly enhance spectral resolution, to explore inter- or intra-molecular interactions in the system, and to decide the sequential order of the events in the system [35,36,37,38].

In our previous study [30], we reported the influence of poly(ethylene glycol) (PEG) molecular weight (*M*n = 400, 1500, and 3400) on the thermal properties of poly-[(R)-3-hydroxybutyrate–*co*–(R)-3-hydroxyhexanoate) (PHBHx, HHx = 6.9 mol %)/PEG blends thin film. We found that PEG with a lower molecular weight (*M*n = 400) was completely miscible with PHBHx while PEG 1500 and 3400 were only partially miscible. We also discovered that a new amorphous band at 1744 cm^−1^, which corresponds to the amorphous mixture of PHBHx and PEG 400, was exhibited only in the 70/30 PHBHx/PEG 400 blend thin film. The possible mixing behavior of a higher molecular weight PEG blended with PHBHx is now explored. In this study, IR mapping was performed to examine the existence of domains in films of PHBHx/PEG (*M*n = 3400) blends. Hyperspectral imaging study provides the localized spatial distributions of multicomponent and multiphase samples [39,40]. Each chemical IR images of PHBHx/PEG with different blending ratios were obtained with different temperatures. The effect of PEG incorporation on PHBHx copolymers was also investigated using temperature-dependent FTIR spectra and 2D-COS.

## 2. Materials and Methods

Biodegradable PHBHx (HHx = 6.9 mol %) copolymer was supplied by the Procter & Gamble Company (Cincinnati, OH, USA) and was purified using process reported previously [30]. PEG (*M*n = 3400) purchased from Sigma-Aldrich Co., Ltd. (St. Louis, MO, USA) was used without further purification.

Blend samples of PHBHx (HHx = 6.9 mol %) with different blending ratios of PEG (*M*n = 3400) were prepared by dissolving them together in chloroform. The blending ratios of PHBHx/PEG blends were 70/30, 60/40 and 50/50. Then PHBHx/PEG blend solution was spun onto a Pt-coated silicon wafer at 1500 rpm for 30 s and subsequently placed under vacuum at 60 °C for 4 h to completely evaporate the residual solvent. The thickness of spin-coated film of PHBHx/PEG blends (70/30, 60/40 and 50/50) was near 5 μm (see the SEM image of cross section of spin-coated film of PHBHx/PEG blend shown in Appendix A).

FTIR spectra and chemical images of PHBHx/PEG blends with increasing temperature (from 30 to 160 °C, interval 10 min) were measured by using a Bruker Vertex 80/v FTIR spectrometer equipped with a Hyperion3000 at the 12D IRS beamline of the Pohang Accelerator Laboratory (PAL). They were conducted with 4 cm^−1^ resolution using a 64 × 64 MCT focal plane array (FPA) detector in ATR mode with a 15× objective. Data were analyzed using Bruker OPUS 7.0 imaging software. FTIR images displayed using chemical images of C=O stretching region at each temperature.

Before 2D-COS analysis for each set of temperature-dependent FTIR spectra of PHBHx/PEG blend films, which were extracted from each chemical image of PHBHx/PEG blends, we performed the baseline correction with PLS_Toolbox 8.7 software (Eigenvector Research, Inc., Wenatchee, WA, USA) for MATLAB R2018b (The Mathworks Inc., Natick, MA, USA). Synchronous and asynchronous 2D correlation FTIR spectra were obtained using MATLAB software. The red and blue lines in 2D correlation spectra displayed positive and negative cross peaks, respectively.

## 3. Results and Discussion

In this study, we focused on the C=O stretching region (1850–1650 cm^−1^) due to the absence of the absorption by PEG to investigate the influence of PEG on PHBHx. Figure 1, Figure 2 and Figure 3 display chemical images of C=O stretching region at each temperature of PHBHx/PEG = 70/30, 60/40 and 50/50 blend films, respectively. As shown in Figure 1, Figure 2 and Figure 3, chemical images of PHBHx/PEG = 70/30, 60/40 and 50/50 blends dramatically changed around 60–80 °C and 130–140 °C. It is in good agreement with the DSC results from our previous study [30]. The change at 60–80 °C is related to the *T*_m_ of PEG while the change at 130–140 °C is related to the *T*_m_ of PHBHx. With the increasing PEG content, chemical images changed near the *T*_m_ of PEG at 50 °C. This observation means that the PHBHx and PEG in these samples have different thermal properties. In other words, they are not fully mixed, in contrast to our previously study [30].

To better understand the thermal behavior of each domain, we analyzed FTIR spectra extracted at different positions in the chemical images. Position A (lowest part) and B (highest part) on the chemical image have been marked as filled circles. Figure 4 displays the temperature-dependent FTIR spectra extracted at parts A and B in Figure 1, Figure 2 and Figure 3. The crystalline C=O stretching band at 1722 (or 1725) cm^−1^ decreases with increasing temperature, while the amorphous C=O stretching band at 1735 cm^−1^ increases with temperature. The shift of carbonyl stretching vibration from 1722 cm^−1^ to a higher wavenumber around 1735 cm^−1^ for the amorphous component is a reflection of the reduced hydrogen bond interactions compared to the crystalline phase [41,42]. In the temperature-dependent FTIR spectra of parts A and B in PHBHx/PEG = 70/30 blend films (as shown in Figure 4A,B), the tendency of intensity changes of two bands at 1722 and 1735 cm^−1^ were similar, even though they had different absorbance values. This result probably means that differences in the chemical image shown in Figure 1 were resulting primarily from thickness variations. In the Figure 4C–F, the intensity changes of two bands at 1722 (or 1725) and 1735 cm^−1^ as a function of temperature are clearly different in parts A and B. Particularly, in the temperature-dependent FTIR spectra of part B in PHBHx/PEG = 50/50 blend film, a band at 1725 cm^−1^ was shifted to lower wavenumber (around 1722 cm^−1^) with increasing temperature. This observation indicates that PEG plays an important role in changing the kinetics and degrees of PHBHx crystallization. In other words, PEG (*M*n = 3400) might have influenced the PHBHx crystallization behavior, because PEG and PHBHx were not fully mixed. Zhao et al., reported that the DSC results of the PHB/PEG blend showed that the degree of crystallinity of PHB changed with different PEG contents [43]. This is in good agreement with our results. As shown in Figure 4E, however, intensity changes of crystalline C=O stretching band at part A in PHBHx/PEG = 50/50 blends are not greater than the corresponding amorphous C=O stretching band. This result indicates that PEG may have more influence on the amorphous C=O stretching mode than the crystalline C=O stretching mode.

To furthermore understand PEG’s effect on PHBHx crystallization in depth, 2D-COS was applied to the temperature-dependent FTIR spectra of PHBHx/PEG = 70/30, 60/40 and 50/50 blend films. Figure 5, Figure 6 and Figure 7 are the 2D correlation spectra obtained from Figure 4.

The synchronous 2D correlation spectra at parts A and B of PHBHx/PEG = 70/30 are quite similar as shown in Figure 5A,C. Two cross peaks at (1723, 1738) and (1712, 1723) cm^−1^ were observed in their corresponding asynchronous 2D correlation spectra as shown in Figure 5B,D. Three bands at 1738, 1723 and 1712 cm^−1^ correspond to amorphous, less-ordered crystalline and well-ordered crystalline C=O stretching modes. This result indicates that differences in chemical images (shown in Figure 1) reflect thickness variation, which is in good agreement with the results in Figure 4A,B.

However, the asynchronous 2D correlation spectra of PHBHx/PEG = 60/40 shown in Figure 6B,D are very different. There is one cross peak at (1723, 1738) cm^−1^ in Figure 6B and at (1715, 1723) cm^−1^ in Figure 6D. The variations of two bands corresponding to the less-ordered crystalline and amorphous C=O stretching modes appeared at part A, while those for two crystalline (well-ordered and less-ordered crystalline) bands appeared at part B in the PHBHx/PEG = 60/40 blend.

In the Figure 7B,D, two cross peaks at (1738, 1748) and (1720, 1738) cm^−1^ and one cross peak at (1720, 1736) cm^−1^ appeared, respectively. A new amorphous band at 1748 cm^−1^ was observed at part A in the PHBHx/PEG = 50/50 blend, which is assigned to the amorphous mixture of PHBHx and PEG [30]. It confirms that with increasing PEG content, partially miscible structures may appear in the PHBHx/PEG = 50/50 blend.

While PEG and PHBHx are only partially miscible, the presence of PEG significantly affects the crystallization behavior of PHBHx, especially the amorphous part in PHBHx. As the mixing of PEG, which acts as a plasticizer, will substantially decrease the *T*_g_ and increase the molecular mobility of PHBHx, we expect the crystallization kinetics will be affected. This effect leads to different thermal behavior of PHBHx copolymers.

## 4. Conclusions

The chemical IR images of PHBHx/PEG blends during the heating process were measured to investigate the existence of domains in films of PHBHx/PEG (high molecular weight) blends. In the chemical images, domains were observed in PHBHx/PEG = 60/40 and 50/50 blends. With the increase in PEG content, a new band at 1748 cm^−1^ assignable to the amorphous partial mixture of PHBHx and PEG appeared. As the mixing of PEG, which acts as a plasticizer, will substantially decrease the *T*_g_ and increase the molecular mobility of PHBHx, we expect the crystallization kinetics will be affected. The results of temperature-dependent FTIR spectra in the domains of the PHBHx/PEG blends revealed that PEG plays an important role in changing the kinetics and degree of PHBHx crystallization. Therefore, PEG can have an effect on changing the thermal behavior of PHBHx copolymers.

## Figures and Tables

**Figure 1 polymers-11-00507-f001:**
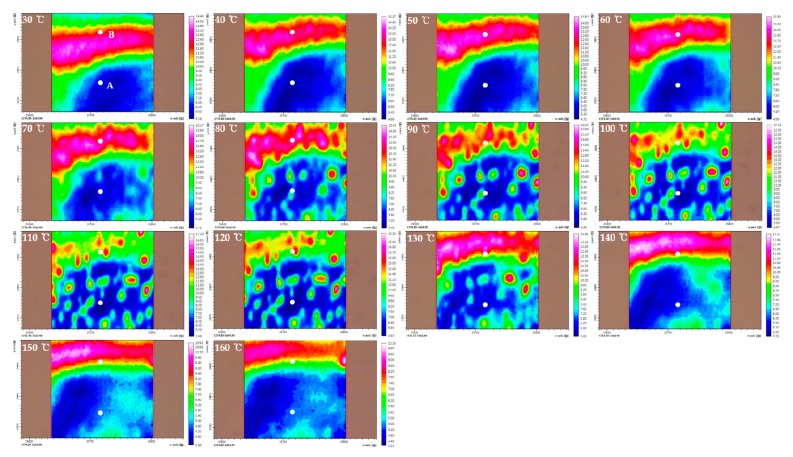
Chemical images of C=O stretching region at each temperature of poly-[(R)-3-hydroxybutyrate-*co*-(R)-3-hydroxyhexanoates] (PHBHx)/poly(ethylene glycol) (PEG) = 70/30 blend film.

**Figure 2 polymers-11-00507-f002:**
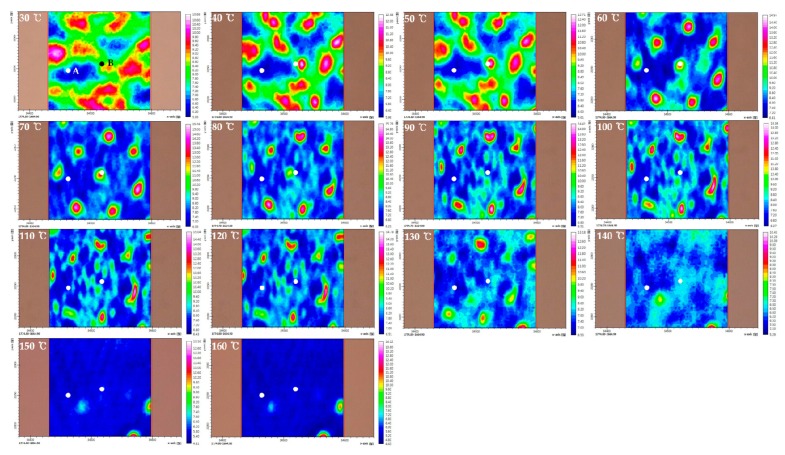
Chemical images of C=O stretching region at each temperature of PHBHx/PEG = 60/40 blend film.

**Figure 3 polymers-11-00507-f003:**
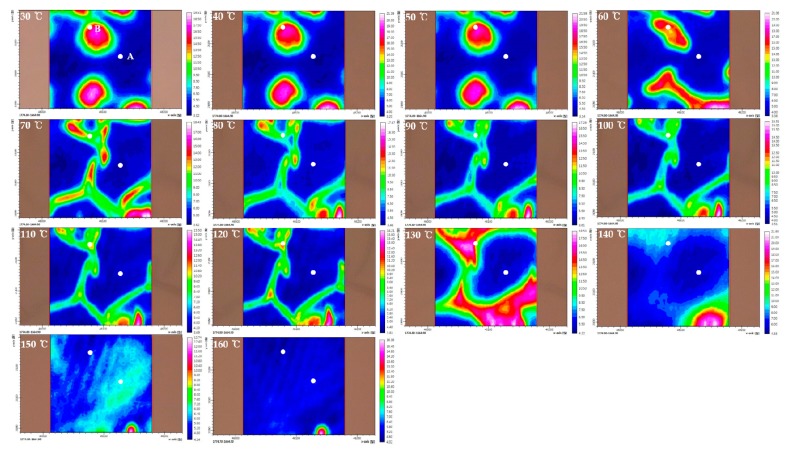
Chemical images of C=O stretching region at each temperature of PHBHx/PEG = 50/50 blend film.

**Figure 4 polymers-11-00507-f004:**
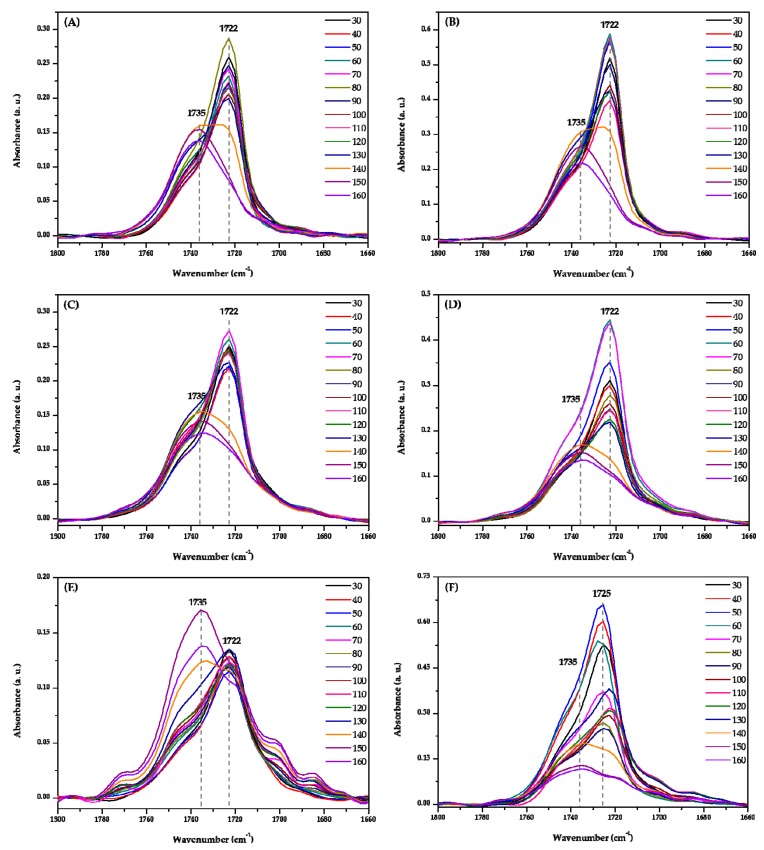
Temperature-dependent FTIR spectra extracted at parts (**A**,**C**,**E**) A and (**B**,**D**,**F**) B in Figure 1, Figure 2 and Figure 3, respectively.

**Figure 5 polymers-11-00507-f005:**
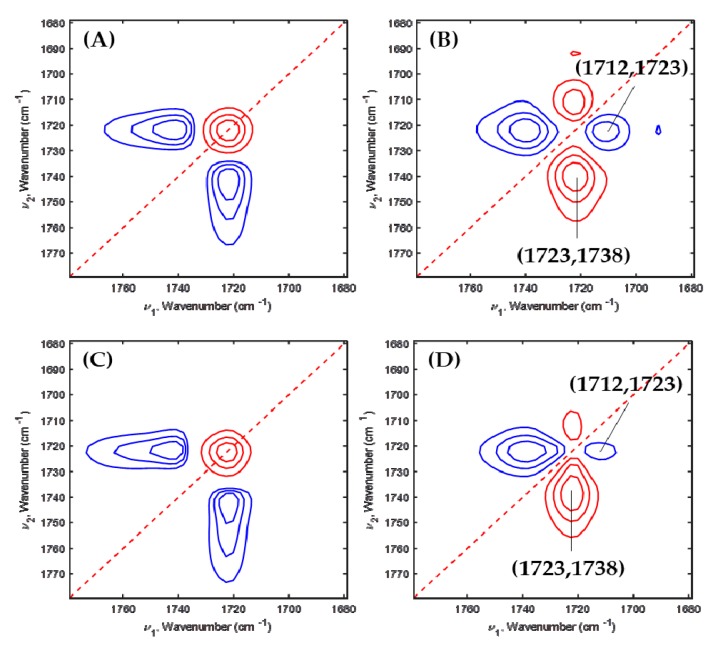
Synchronous (**A**,**C**) and asynchronous (**B**,**D**) 2D correlation spectra obtained from Figure 4A,B.

**Figure 6 polymers-11-00507-f006:**
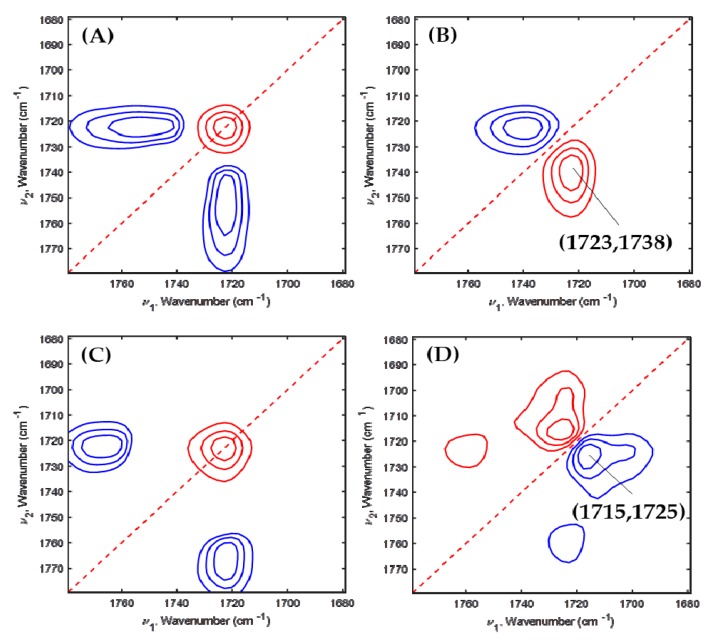
Synchronous (**A**,**C**) and asynchronous (**B**,**D**) 2D correlation spectra obtained from Figure 4C,D.

**Figure 7 polymers-11-00507-f007:**
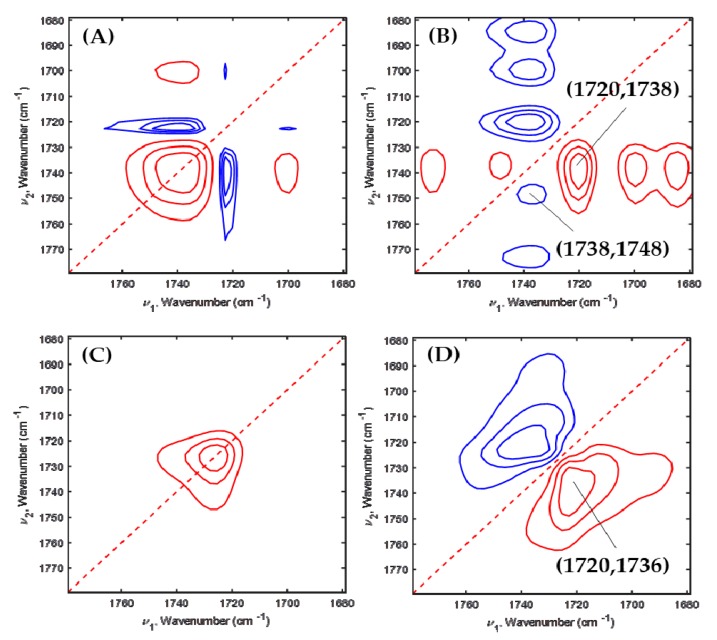
Synchronous (**A**,**C**) and asynchronous (**B**,**D**) 2D correlation spectra obtained from Figure 4E,F.

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
