# Peer review of "Studies on Chemical IR Images of Poly(hydroxybutyrate–co–hydroxyhexanoate)/Poly(ethylene glycol) Blends and Two-Dimensional Correlation Spectroscopy"

_polymers, 2019, doi:10.3390/polym11030507_

Round 1
Reviewer 1 Report
This is an interesting reseach and should have some impact. I recommend the publication this manuscript after mino revision.
The authors should have more detailed introduction to IR imaging of polymers and should cite important papers. For examples, papers by Kazarian et aal. and those by Siesler et al.,,,
In Figure 4(F) a band near 1722 cm-1 showed a lower wavenumber shift. What is the cause of the shift?
The authors should compare more the present results with those by other research groups.
4. Conclusion is a bit weak. The authors shoud strengthen it significantly.
Author Response
All the replied are written in boldface.
We appreciate reviewer’s comments on our manuscript, which have made us to significantly revise the manuscript.
Response on comments of Reviewer 1
This is an interesting research and should have some impact. I recommend the publication this manuscript after minor revision.
The authors should have more detailed introduction to IR imaging of polymers and should cite important papers. For examples, papers by Kazarian et al. and those by Siesler et al.,,,
According to Reviewer’s comments, we added this following sentence and cited two references accordingly in the revised manuscript.
“Hyperspectral imaging study provides the localized spatial distributions of multicomponent and multiphase samples[39,40].”
This content appears at the Introduction (in page 2) of the revised manuscript.
In Figure 4(F) a band near 1722 cm-1 showed a lower wavenumber shift. What is the cause of the shift?
According to Reviewer’s comments, we added this following sentence in the revised manuscript.
“The shift of carbonyl stretching vibration from 1722 cm-1 to a higher wavenumber around 1735 cm-1 for the amorphous component is a reflection of the reduced hydrogen bonding interaction compared to crystalline phase[41,42].”
“Especially, in the temperature-dependent FTIR spectra of part B in PHBHx/PEG=50/50 blend film, a band at 1725 cm-1 was shifted to lower wavenumber (around 1722 cm-1) with increasing temperature. This observation indicates that PEG plays an important role in changing the kinetics and degrees of PHBHx crystallization.”
These contents appear at the Results and Discussion (in page 4) of the revised manuscript.
The authors should compare more the present results with those by other research groups.
According to Reviewer’s comments, we added this following sentence in the revised manuscript.
“The shift of carbonyl stretching vibration from 1722 cm-1 to a higher wavenumber around 1735 cm-1 for the amorphous component is a reflection of the reduced hydrogen bonding interaction compared to crystalline phase[41,42].”
“Zhao et al., reported that DSC results of PHB/PEG blend showed that degree of crystallinity of PHB changed with different PEG contents[43]. It is in good agreement with our results.”
These contents appear at the Results and Discussion (in page 4) of the revised manuscript.
Conclusion is a bit weak. The authors should strengthen it significantly.
According to Reviewer’s comments, we added this following sentence in the revised manuscript.
“With the increase in PEG content, a new band at 1748 cm-1 assignable to the amorphous partial mixture of PHBHx and PEG appeared. As the mixing of PEG, which acts as a plasticizer, will substantially decrease the Tg and increase the molecular mobility of PHBHx, we expect the crystallization kinetics will be affected.”
This content appears at the Conclusions (in page 7) of the revised manuscript.

Reviewer 2 Report
The present contribution investigates a biodegradable polymer blend by FTIR-imaging and 2D-COS analysis of the resulting hyperspectral data sets. The main conclusion of the study is that the poly(ethylene glycol) component influences the crystallization kinetics and the thermal behavior of the co-polyester with which it is blended. The study illustrates the potential of FTIR imaging and 2D-COS for the characterization of polymer blends and is expected to be of interest for a wide readership. The experiments are suitably designed and the conclusions are sound and adequately supported by the reported results. Comprehensive reference to the existing literature is given.
I have the following comments:
1. The IR images in Figs. 1 and 2 need to be improved. Scales are unreadable. It is advisable to draw a marker for size. The last two images of Fig. 1 (150°C and 160°C) give the (wrong) impression that the carbonyl groups are no longer present in the sampling area. This is a scale effect that should be avoided or, at least, commented.
2. Application of 2D-COS analysis to hyperspectral data is an interesting and potentially useful approach. However, the underlining principles are not obvious. In particular, difficulties may arise when trying to extend the concept of perturbation variable to the space coordinates of an image. A discussion of the subject would be timely and useful.
3. English language needs to be improved in several places (for instance, line 31: … is as low 9%; line 53: … and PEG 400 observed, was observed only …; lines 103 – 104: The position A (lowest part) and B (highest part) have marked on the chemical image …; line 131: C=O stretching mods; etc.
In summary, in my opinion the present study can be considered for publication in POLYMERS provided the authors suitably address the above comments.
Author Response
All the replied are written in boldface.
We appreciate reviewer’s comments on our manuscript, which have made us to significantly revise the manuscript.
Response on comments of Reviewer 2
The present contribution investigates a biodegradable polymer blend by FTIR-imaging and 2D-COS analysis of the resulting hyperspectral data sets. The main conclusion of the study is that the poly(ethylene glycol) component influences the crystallization kinetics and the thermal behavior of the co-polyester with which it is blended. The study illustrates the potential of FTIR imaging and 2D-COS for the characterization of polymer blends and is expected to be of interest for a wide readership. The experiments are suitably designed and the conclusions are sound and adequately supported by the reported results. Comprehensive reference to the existing literature is given.
The IR images in Figs. 1 and 2 need to be improved. Scales are unreadable. It is advisable to draw a marker for size.
According to Reviewer’s comments, we changed Figures 1, 2, and 3 in the revised manuscript.
The last two images of Fig. 1 (150°C and 160°C) give the (wrong) impression that the carbonyl groups are no longer present in the sampling area. This is a scale effect that should be avoided or, at least, commented.
As you can see Figures 4(a) and 4(b), carbonyl groups are obviously appeared. In the Figure 3, the carbonyl group appears to be no longer present in the sampling area. This observation indicates that the spectral intensities of the carbonyl groups in almost all regions are almost the same.
Application of 2D-COS analysis to hyperspectral data is an interesting and potentially useful approach. However, the underlining principles are not obvious. In particular, difficulties may arise when trying to extend the concept of perturbation variable to the space coordinates of an image. A discussion of the subject would be timely and useful.
According to Reviewer’s comments, we added this following sentence in the revised manuscript.
“2D-COS can sort out very important and interesting information in the system, which is sometimes scarcely detected in conventional 1D spectral analysis. It has many powerful advantages to greatly enhance spectral resolution, to explore inter- or intra-molecular interaction in the system, and to decide the sequential order of the events in the system[35–38].”
This content appears at the Introduction (in page 2) of the revised manuscript.
English language needs to be improved in several places (for instance, line 31: … is as low 9%; line 53: … and PEG 400 observed, was observed only …; lines 103 – 104: The position A (lowest part) and B (highest part) have marked on the chemical image …; line 131: C=O stretching mods; etc.
According to Reviewer’s comments, we carefully revised our manuscript. We also corrected several places pointed out by reviewer in the revised manuscript.
“… is as low 9%, …” changed to “… is as low as 9%, …”
This content appears at the Introduction (in page 1) of the revised manuscript.
“PEG 400 observed, was observed only….” changed to “PEG 400, was exhibited only…”
This content appears at the Introduction (in page 2) of the revised manuscript.
“The position A (lowest part) and B (highest part) have marked on the chemical image as filled circles.” changed to “The position A (lowest part) and B (highest part) on the chemical image have marked as filled circles.”
This content appears at the Introduction (in page 4) of the revised manuscript.
“C=O stretching mods” changed to “C=O stretching modes”
This content appears at the Results and Discussion (in page 4) of the revised manuscript.
In summary, in my opinion the present study can be considered for publication in POLYMERS provided the authors suitably address the above comments.
Reviewer 3 Report
In this manuscript, authors investigated the effect of the high molecular weight (Mn=3400) PEG blended in the films of PHBHx with different ratios of PEG by using chemical FTIR imaging. The manuscript is well-organized, and conclusions seems in agreement with the obtained data. Therefore, I recommend this manuscript to be published after minor revisions as suggested belows:
1. Have authors checked the optical image of the PHBHx/PEG blend films? Is there any difference between the optical image and chemical FTIR image?
2. In addition, what kinds of new information did author obtain from the FTIR imaging?
Author Response
All the replied are written in boldface.
We appreciate reviewer’s comments on our manuscript, which have made us to significantly revise the manuscript.
Response on comments of Reviewer 3
In this manuscript, authors investigated the effect of the high molecular weight (Mn=3400) PEG blended in the films of PHBHx with different ratios of PEG by using chemical FTIR imaging. The manuscript is well-organized, and conclusions seems in agreement with the obtained data. Therefore, I recommend this manuscript to be published after minor revisions as suggested bellows:
Have authors checked the optical image of the PHBHx/PEG blend films? Is there any difference between the optical image and chemical FTIR image?
Yes, we already measured the optical image of the PHBHx/PEG blend films and also compared the optical image with chemical FTIR image. As shown in Figures A(a), A(c) and A(e), PHBHx/PEG blend film looks almost the same. However, each chemical images of PHBHx/PEG blend films are different. We expected that each mixed film in the FTIR image would have the domain or thickness of the film.
Figure A. The optical images (a, c, and e) and the FTIR images (b, d, and f) of PHBHx/PEG=70/30 (a and b), 60/40 (c and d), and 50/50 (e and f) blend films at 30 °C.
2. In addition, what kinds of new information did author obtain from the FTIR imaging?
As you can see the Figure A, we cannot get any information only from optical images. However, chemical information can be obtained clearly from FTIR images. From the FTIR images, PHBHx/PEG = 60/40 and 50/50 blends have domains and different thermal behaviors. Therefore, we conclude that PEG can affect the thermal behavior of PHBHx copolymers.

Round 2
Reviewer 1 Report
The authors prepared a strong revised version based on the comments from the reviewer. Now I can recommend the publication of this version in Polymers
Reviewer 2 Report
The authors have addressed thel reviewer comments
Reviewer 3 Report
The manuscript has been improved largely, it can be published as it is.